# Investigation of an Outbreak of Equine Herpesvirus-1 Myeloencephalopathy in a Population of Aged Working Equids

**DOI:** 10.3390/v16121963

**Published:** 2024-12-21

**Authors:** Nicola Pusterla, Kaila Lawton, Samantha Barnum, Kelly Ross, Kris Purcell

**Affiliations:** 1Department of Medicine and Epidemiology, School of Veterinary Medicine, University of California, Davis, CA 95616, USA; kolawton@ucdavis.edu (K.L.); smmapes@ucdavis.edu (S.B.); 2Camp Richardson Corral, South Lake Tahoe, CA 96150, USA; qrkr@sbcglobal.net; 3Carson Valley Large Animal Clinic, Gardnerville, NV 89460, USA; scubavets@aol.com

**Keywords:** EHV-1, equine herpesvirus myeloencephalopathy, outbreak, aged horses, fatality rate

## Abstract

The objective of this study was to describe an outbreak of equine herpesvirus-1 myeloencephalopathy (EHM) in a population of aged equids. The outbreak was linked to the introduction of five healthy non-resident horses 15 days prior to the first case of acute recumbency. This fulminant EHM outbreak was predisposed by the grouping of the 33 unvaccinated animals in two large pens with shared water and feed troughs. Fourteen horses (42.4%) developed neurological deficits within the first week of the outbreak. Four additional equids developed fever and respiratory signs (EHV-1 infection), while fifteen horses remained healthy. EHM was supported by the detection of EHV-1 N_752_ in blood (*n* = 11) and/or nasal secretions (9). Three out of four equids with EHV-1 infection and two out of fifteen healthy horses tested qPCR-positive for EHV-1. All animals were managed in the field. EHM and EHV-1 equids were treated with a combination of antiherpetic, anti-inflammatory, and antithrombotic drugs. Six out of fourteen EHM horses (42.9%) were euthanized because of recumbence and the inability to stand with assistance or vestibular signs. Anti-EHV-1 total IgG and IgG 4/7 levels in acute serum samples showed no significant difference amongst the three disease groups (*p* > 0.05); however, antibody levels rose significantly between acute and convalescent serum samples for EHM (*p* = 0.0001) and EHV-1 equids (*p* = 0.02). This outbreak highlights a very high EHM attack and fatality rate in a population of aged equids and rapid spread of EHV-1, as the population shared common pens and feeding practices. The outbreak also showed that EHM cases can be managed in the field when referral to a hospital is not an option.

## 1. Introduction

Equine herpesvirus-1 myeloencephalopathy (EHM) continues to impact the equine industry in North America and Europe through high morbidity and fatality, costs of treatment, and cancellation of equestrian events [1]. While news and media often report on large EHM outbreaks at shows due to their multi-state or multi-country involvement [2,3,4], smaller and local outbreaks are as relevant in order to study and understand their epidemiology. EHM is considered a multifactorial disease of adult horses and a late-term complication of an EHV-1 infection [5]. Unfortunately, EHV-1 infections can remain unnoticed for days until the first neurologically infected horse is noticed and a diagnosis of EHM is supported through the detection of EHV-1 in blood and/or nasal secretions [4,6]. At that time, the virus has silently spread amongst herd mates, leading to high morbidity. Various risk factors have been associated with the development of EHM, including but not restricted to stress, transportation, size of the horse population, sex, age, breed, number of classes entered during a show, vaccination, poor barn ventilation, and biosecurity risks [2,4,5,7,8,9,10]. However, it is important to keep in mind that risk factors associated with EHM are not always consistent and can be contradictory, as each outbreak is variable regarding the virus, the affected horse population, and husbandry practices. Here, we report on an EHM outbreak in an older population of equids. The outbreak was characterized by a wide spread of the virus, a high morbidity and fatality rate, and practical management of the outbreak at the ranch due to financial constraints.

## 2. Materials and Methods

The outbreak happened in the summer of 2024 at a horseback riding operation located in the Sierra Nevada Mountains of California. The population was composed of 28 resident equids and 5 outside horses, which were brought to the ranch 15 days prior to the first neurological case. The 5 outside horses were kept separated from the remaining horses for 9 days. All equids were kept in two large pens with shared feed and water troughs. The animals were vaccinated against core antigens, including equine eastern and western encephalitis virus, west Nile virus, and tetanus toxoid, in the spring of 2024, but they had not been vaccinated against EHV-1, EHV-4, or equine influenza virus. On June the 3^rd^, a 23-year-old Quarter horse gelding developed acute recumbency and was euthanized with no necropsy performed. On the same day, a 20-year-old draft horse developed fever (104.0 °F), and blood and nasal secretions tested qPCR-positive for EHV-1 (N_752_). Thereafter, clinical data and individual medical treatments were recorded in real-time from the onset of the outbreak (day 0) up to the lifting of the quarantine (day 35). Sample collections (blood and nasal secretions) were performed by the attending veterinarian at various time points (days 2, 7, 15, and 23) on the majority of the study horses (27 to 31 horses).

Whole blood was collected in evacuated tubes (BD Vacutainer^®^, Franklin Lakes, NJ, USA), and nasal secretions were collected using 6-inch-long rayon-tipped swabs (Puritan^®^ Sterile Rayon Tipped Applicators, Guilford, ME, USA). Nucleic acid extraction from whole blood and nasal secretions was performed 24 h post-collection using an automated nucleic acid extraction system (QIAcube HT, Qiagen, Valencia, CA, USA) according to the manufacturer’s recommendations. All samples were assayed for the presence of the equine glyceraldehyde-3-phosphate dehydrogenase (eGAPDH) gene, the glycoprotein B (gB), and the polymerase (ORF 30) gene of EHV-1 using previously reported real-time TaqMan PCR assays [11,12].

A 1 mL aliquot of serum collected during the acute and convalescent period of the outbreak was kept refrigerated at −80 °C until completion of the study, at which time the samples were sent in one batch to the Animal Health Diagnostic Center at Cornell University for the testing of anti-EHV-1 total IgG and IgG 4/7 using a multiplex assay, as previously reported [13]. This EHV-1 risk evaluation assay quantitates special subsets of antibodies against EHV-1 in serum from equids for the purpose of assessing susceptibility to development of infection and clinical disease or determining protection. Based on the quantitative values of the serological results for IgG and IgG 4/7, equids were assigned a risk of developing EHV-1 disease (high risk: IgG < 3000 MFI and IgG 4/7 < 400 MFI; moderate risk: IgG < 3000 MFI and IgG 4/7 ≥ 400 MFI or IgG ≥ 3000 and IgG 4/7 < 400; low risk: IgG > 3000 MFI and IgG 4/7 > 400 MFI; very low to no risk: IgG > 12,000 MFI and IgG 4/7 > 10,000 MFI).

## 3. Results

The population of equids was composed of 32 horses and 1 mule ranging in age from 7 to 30 years (median 20 years). There were 8 mares and 25 geldings. Amongst the horses there were 14 draft horses, 9 quarter horses, 4 paint horses, 2 Tennessee walking horses, 1 appaloosa horse, 1 thoroughbred, and 1 mustang.

Abnormal clinical signs developed during the first week of the outbreak in 18 equids, while 15 horses (healthy cases) did not develop any abnormal clinical signs during the entire observation period (Table 1). Fourteen horses developed acute onset of neurological deficits, including ataxia, weakness, vestibular signs, urinary incontinence, and recumbency (EHM cases). Three horses and one mule developed fever, lethargy, nasal discharge, and/or coughing (EHV-1 cases). Six neurological horses with either recumbency or acute onset of vestibular signs were euthanized. Necropsy was not performed on any of the euthanized horses.

All 12 EHM cases for which biological samples were available tested EHV-1 qPCR-positive in blood (n = 11) and/or nasal secretions (n = 9). For the four EHV-1 cases, three horses tested qPCR-positive in blood and nasal secretions, while one animal tested negative on both samples. Most of the healthy horses (n = 13) tested EHV-1 qPCR-negative in blood and nasal secretions throughout the outbreak, while one horse was positive only in blood and another horse was positive in blood and nasal secretions. During the first sample collection (48 h after onset of outbreak, i.e., day 2) 15/31 and 13/31 horses tested qPCR-positive for EHV-1 in blood and in nasal secretions, respectively (Table 2). The EHV-1 viral loads at the gDNA level in blood ranged from 84 to 6.1 × 10^4^ gB gene copies/million cells (median 2.8 × 10^3^ gB gene copies/million cells), and the viral load in nasal secretions ranged from 81 to 2.5 × 10^6^ gB gene copies/million cells (median 5.5 × 10^4^ gB gene copies/million cells). At the second collection time point (day 7), five and three horses tested qPCR-positive for EHV-1 in blood and in nasal secretions, respectively. The EHV-1 viral loads in blood ranged from 299 to 1.5 × 10^4^ gB gene copies/million cells (median 468 gB gene copies/million cells). The EHV-1 viral loads in nasal secretions ranged from 2033 to 4.1 × 10^4^ gB gene copies/million cells (median 1.1 × 10^4^ gB gene copies/million cells). On day 15 of the outbreak, only one horse tested qPCR-positive for EHV-1 in blood (viral load 156 gB gene copies/million cells), and another horse tested qPCR-positive in nasal secretions (viral load 496 gB gene copies/million cells). By day 23 of the outbreak, all horses tested EHV-1 qPCR-negative in blood and in nasal secretions. All EHV-1 qPCR-positive blood and nasal secretions were of the non-neuropathogenic genotype (N_752_).

Treatment was left at the discretion of the attending veterinarian and the ranch owner. Horses with neurological signs were treated with a combination of valacyclovir (*n* = 10) for 1 to 13 days (median 6 days), flunixin meglumine (*n* = 9) for 2 to 12 days (median 6.5 days), dexamethasone (*n* = 10) for 3 days, and sodium heparin or enoxaparin sodium (*n* = 4) for 1 to 3 days (median 2 days). The 3 EHV-1 horses with fever and viremia were treated with valacyclovir (7 days), phenylbutazone (5 days), dexamethasone (3 days), and sodium heparin or enoxaparin sodium (3 days). The EHV-1 suspect horse with no viremia was not treated. Furthermore, all diseased horses were supplemented with oral vitamin E. Five of the neurological horses regained full function by the end of the outbreak. Three additional horses were still showing hind limb weakness and difficulty getting up 5 months after the onset of neurological signs. None of the EHV-1 cases developed neurological signs following initiation of treatment.

All 31 horses had detectable anti-EHV-1 antibodies during the acute stage (Table 3; Figure 1). Acute serum anti-EHV-1 total IgG and IgG 4/7 antibody values did not significantly differ amongst the three disease groups (*p* > 0.05; repeated measures ANOVA). However, the total IgG and IgG 4/7 values significantly increased for the EHM (*p* = 0.0001) and EHV-1 (*p* = 0.02) groups between acute and convalescent serum samples. The total IgG and IgG 4/7 values for acute and convalescent serum samples were not significantly different for the healthy horse group (*p* > 0.05). The risk of EHV-1 disease development did not appear to be associated with total IgG and IgG 4/7 levels of the acute serum samples. While eight EMM horses had a high or moderate risk of disease development, four EHM horses had a low risk. A similar pattern was observed for the EHV-1 diseased and healthy horse groups. Furthermore, outcomes did not appear to relate to the antibody levels collected during the acute phase, as the euthanized EHM horses had a moderate risk, while the risk of EHM survivors ranged from high to low (Table 3).

## 4. Discussion

This outbreak offers a unique perspective on clinical and laboratory findings, field treatment options, and disease outcomes in a population of aged horses. The fulminant nature of the outbreak was related to the husbandry practices at the horseback riding operation, with all 33 horses being kept in two adjacent pens with shared water and feeding troughs. While the source of the outbreak has remained unknown, it is possible that the EHV-1 originated from the five non-resident horses, which were brought to the facility 15 days prior to the first index case. This observation is further supported by the lack of clinical disease in the five non-resident horses, a lack of EHV-1 detection at the onset of the outbreak, and a lack of increase in antibody levels between acute and convalescent serum samples. The introduction of non-resident horses into an established population of equids should be considered a risk factor for EHM outbreaks [14,15]. While the transmission of EHV-1 from subclinical shedders to susceptible horses is difficult to prevent, this outbreak highlights the need to isolate non-resident horses for at least 21 days with proper biosecurity measures in place in order to prevent spread. Unfortunately, it is not uncommon for an EHM outbreak to only be recognized at the onset of the first neurologically affected horse [2,3,4]. The time-lag between EHV-1 infection and the development of EHM often allows the virus to spread silently amongst susceptible horses. This scenario has repeatedly been experienced in large EHM outbreaks [2,4] and emphasizes the need to monitor resident and non-resident horses for early clinical signs at times when physical and medical interventions can prevent the devastating outcomes of EHM [12].

It is thought that a combination of host and viral factors determine whether EHM occurs. Older age has been linked to the development of EHM and likely relates to alterations in mucosal and systemic immune responses [16,17]. The susceptibility to EHV-1 infection and development of EHM was evident in the present case series, as the age of the 14 EHM cases ranged from 19 to 26 years (median 21 years of age). While the morbidity (18/33, 54.5%) and mortality rates (6/33, 18.2%) for EHV-1 infections were similar to previously reported outbreaks [5], the EHM attack rate (14/33, 42.4%) and EHM fatality rates were very high (6/14, of 42.9%). It remains unclear how the lack of vaccination against EHV-1 may have impacted this outbreak. Commercially available killed and modified-life vaccines minimally reduce the incidence of clinical disease associated with EHV-1 infection [18]. No definitive consensus has been reached regarding the role of EHV-1 vaccination as a risk factor for EHM. The data are contradictory, with some studies showing protection and greater survival in vaccinated horses [10] and other studies showing a higher risk of EHM development in recently vaccinated horses [2].

The clinical presentation of EHM cases was similar to previous reports and characterized by hind limb ataxia and proprioceptive deficits, weakness, urinary incontinence, vestibular signs, and recumbency consistent with multifocal vasculopathy within the central nervous system [7]. Recumbency with inability to stand with assistance as well as vestibular signs led to the euthanasia of a total of six horses. EHV-1 was detected through qPCR in blood and/or nasal secretions in all EHM cases, in three EHV-1 cases, and in two healthy horses. While subclinically infected horses can shed EHV-1 in nasal secretions, the presence of EHV-1 viremia is a relatively rare event in such horses [19]. It was interesting to observe that one of the four EHV-1 diseased horses tested qPCR-negative in both blood and nasal secretions. This horse was sampled prior to the development of fever, which may explain the qPCR-negative results. However, it is also possible that the fever in this case was unrelated to an EHV-1 infection. Duration and peak viremia have been associated with the development of EHM [16,19]. In the present study, the median viral load in blood was 10-fold higher in EHM cases compared to EHV-1 and healthy cases combined. Previous studies have shown that non-surviving EHM cases have significantly higher viremia levelscompared to surviving EHM cases [20]. In the present study, non-surviving EHM cases displayed a blood median viral load of 21,255 gB gene copies/million cells compared to the blood median viral load of 2426 gB gene copies/million cells in surviving EHM cases. Higher viral loads in blood may be associated with greater disease severity and mortality, as shown for other viral diseases [21].

Few data are available regarding the outcome of early medical intervention in horses with EHM [22]. Anti-inflammatory and antithrombotic drugs may reduce inflammation and prevent further thromboembolic events, thereby reducing the extent of spinal cord damage [5,7,22]. Lowering the viral load in lymphocytes/monocytes during cell-associated viremia using antiherpetic drugs has been one of the strategies to reduce the risk of EHM development in febrile horses. In the present study, none of the three horses with fever and viremia treated with valacyclovir ended up developing EHM, although the number of treated horses is too low to draw any conclusions. The use of valacyclovir in viremic horses had the positive effect of quickly reducing viremia and nasal shedding, as exemplified by the rapid decline of EHV-1 qPCR-positive horses on the second sample collection date (day 7 following onset of the outbreak).

Clinical disease (EHM and EHV-1) significantly increased the serum levels of anti-EHV-1 total IgG and IgG 4/7 compared to healthy horses. However, initial values were not associated with disease form or outcome. Previous work has shown that the concentration of antibodies to EHV-1 measured via serum neutralization assay prior to virus inoculation did not correlate with protection against challenge with EHV-1 [16]. However, a recent study showed that pre-existing mucosal anti-EHV-1 IgG1 and IgG 4/7 antibodies were able to neutralize EHV-1, therefore preventing clinical disease, reducing shedding, and impeding viremia [23]. The measurement of mucosal anti-EHV-1 antibodies during onset of an outbreak may help in the future to assess the risk of EHM development.

As with any observational study, there are various inherent limitations to the present study. First of all, collection of samples was not available for all horses, as two EHM suspected cases (one of them being the index case) were not tested for EHV-1. Furthermore, while all study animals were monitored daily, the clinical information was not always recorded, meaning subtle and transient elevations of rectal temperature may have been missed. Additionally, the treatment of the horses was not standardized and left to the discretion of the attending veterinarian and the ranch owner. Treatment options were impacted by financial limitations and the severity of disease on initial presentation. However, this observational study showed that EHM cases can be treated in the field with positive outcomes.

## 5. Conclusions

This study describes the investigation of an EHM outbreak in a group of 33 older horses used for recreational horseback riding. EHV-1 spread rapidly due to the commingling of the study horses in two pens and the use of shared water and feeding troughs. Age was considered a risk factor for the high EHM attack and fatality rate. Furthermore, higher viral loads in blood were associated with EHM and mortality, while anti-EHV-1 antibody levels in acute serum samples could not predict protection against disease. In hindsight, field outbreaks cannot be prevented, as silent and clinical transmission of EHV-1 is inherent to equine populations, and monitoring and biosecurity protocols are generally minimal outside of at-risk populations. However, the authors believe that two main procedures could have positively impacted the outcome of this outbreak: (i) an appropriate isolation period (21 days) for non-resident horses with daily monitoring and (ii) proper biosecurity protocols and daily monitoring of resident horses to recognize early disease and isolate such horses.

## Figures and Tables

**Figure 1 viruses-16-01963-f001:**
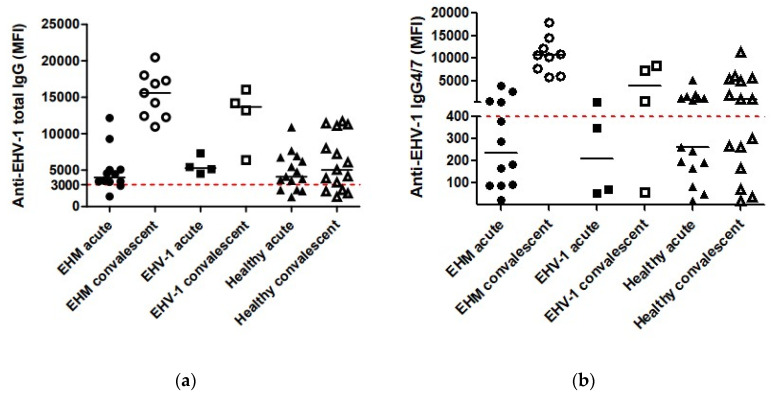
(**a**) Anti-EHV-1 total IgG in acute and convalescent serum samples collected from horses with EHM, EHV-1 infection and healthy horses involved in an outbreak. The median is represented by horizontal bars at each time point. The red dotted line at 3000 MFI represents the protective cut-off for the EHV-1 risk evaluation assay. Antibody levels are expressed as median fluorescence intensity (MFI). (**b**) Anti-EHV-1 IgG 4/7 in acute and convalescent serum samples collected from horses with EHM, EHV-1 infection and healthy horses involved in an outbreak. The median is represented by horizontal bars at each time point. The red dotted line at 400 MFI represents the protective cut-off for the EHV-1 risk evaluation assay. Antibody levels are expressed as median fluorescence intensity (MFI).

**Table 1 viruses-16-01963-t001:** Signalment, clinical, molecular, and serological results of 33 horses involved in an EHM outbreak.

Horse Year/Sex/Breed	Clinical Disease	qPCR		Serology (Acute Serum)			Outcome
		Nasal Shedding	Viremia	IgG (MFI)	IgG 4/7 (MFI)	EHV-1 Disease Risk	
23/G/Quarter horse	EHM	NA	NA	NA	NA	NA	Euthanized
19/G/Quarter horse	EHM	NA	NA	NA	NA	NA	Euthanized
23/F/Quarter horse	EHM	-	+	3536	182	Moderate	Euthanized
21/F/Quarter horse	EHM	+	+	3376	92	Moderate	Euthanized
21/F/Paint horse	EHM	-	+	5024	378	Moderate	Euthanized
26/G/Draft horse	EHM	+	+	4442	284	Moderate	Euthanized
20/G/Draft horse	EHM	+	+	1381	21	High	Alive
20/G/Draft horse	EHM	+	+	2845	86	High	Alive
19/G/Appaloosa	EHM	+	+	3447	166	Moderate	Alive
20/G/Paint horse	EHM	+	+	3445	88	Moderate	Alive
21/F/Thoroughbred	EHM	+	+	5083	1090	Low	Alive
25/G/Quarter horse	EHM	+	+	4547	431	Low	Alive
23/G/Quarter horse	EHM	+	-	12,153	3857	Low	Alive
20/G/Quarter horse	EHM	-	+	9281	2706	Low	Alive
10/G/Quarter horse	EHV-1	+	+	5125	409	Low	Alive
23/F/Mule	EHV-1	-	-	4500	71	Moderate	Alive
7/F/Draft horse	EHV-1	+	+	7307	1679	Low	Alive
25/G/Quarter horse	EHV-1	+	+	5427	345	Moderate	Alive
21/G/Draft horse	Healthy	-	+	6186	1168	Low	Alive
15/G/Draft horse	Healthy	+	+	3639	164	Moderate	Alive
10/G/Draft horse *	Healthy	-	-	6915	1090	Low	Alive
21/G/Draft horse *	Healthy	-	-	3539	260	Moderate	Alive
15/F/Draft horse *	Healthy	-	-	10,874	5249	Low	Alive
15/G/Draft horse *	Healthy	-	-	7644	1980	Low	Alive
16/G/Paint horse *	Healthy	-	-	5421	715	Low	Alive
15/G/TWH	Healthy	-	-	2244	51	High	Alive
24/G/Draft horse	Healthy	-	-	4087	191	Moderate	Alive
23/G/Draft horse	Healthy	-	-	3758	197	Moderate	Alive
23/G/Draft horse	Healthy	-	-	4755	241	Moderate	Alive
15/G/Paint horse	Healthy	-	-	2255	82	High	Alive
30/G/Mustang	Healthy	-	-	6722	1083	Low	Alive
16/G/Draft horse	Healthy	-	-	2084	48	High	Alive
20/G/TWH	Healthy	-	-	1283	16	High	Alive

G = gelding; F = female; TWH = Tennessee walking horse; NA = not available; * = non-resident horse; EHM = equine herpesvirus myeloencephalopathy (neurological signs including ataxia, weakness, proprioceptive deficits, vestibular signs, urinary incontinence, recumbency); EHV-1 (fever, lethargy, nasal discharge, coughing); MFI: median fluorescence intensity.

**Table 2 viruses-16-01963-t002:** EHV-1 qPCR viral load results in blood and nasal secretions of 31 horses involved in an EHM outbreak. The range and the median are expressed as the number of EHV-1 gB target genes per million cells.

Day	qPCR Results of Blood		qPCR Results of Nasal Secretions	
	Pos/Neg	Range (Median)	Pos/Neg	Range (Median)
2	15/16	84–6.1 × 10^4^ (2818)	13/16	81–2.5 × 10^6^ (5.5 × 10^4^)
7	5/23	299–1.5 × 10^4^ (468)	3/25	2033–4.1 × 10^4^ (1.1 × 10^4^)
15	1/27	156	1/27	496
23	0/27	0	0/27	0

**Table 3 viruses-16-01963-t003:** Anti-EHV-1 total IgG and IgG4/7 in serum of horses involved in an EHM outbreak. The horses were grouped by clinical presentation into EHM, EHV-1, and healthy horses. Based on the serological values for acute and convalescent serum samples, the horses were assigned a risk score [13].

Risk	EHM		EHV-1		Healthy	
	Acute	Convalescent	Acute	Convalescent	Acute	Convalescent
High	2	0	0	0	4	4
Moderate	6	3	2	0	5	4
Low	4	6	2	3	6	7
No	0	0	0	1	0	0

High risk: IgG < 3000 MFI and IgG 4/7 < 400 MFI; moderate risk: IgG < 3000 MFI and IgG 4/7 ≥ 400 MFI or IgG ≥ 3000 and IgG 4/7 < 400; low risk: IgG > 3000 MFI and IgG 4/7 > 400 MFI; very low to no risk: IgG > 12,000 MFI and IgG 4/7 > 10,000 MFI.

## Data Availability

Data are contained within the article.

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
