# Peer review of "Investigation of an Outbreak of Equine Herpesvirus-1 Myeloencephalopathy in a Population of Aged Working Equids"

_viruses, 2024, doi:10.3390/v16121963_

Round 1
Reviewer 1 Report
Comments and Suggestions for Authors
This well-written description of an EHV-1 outbreak illustrates the rapid onset of EHV-1 induced disease in a group of 32 unvaccinated, aged horses and one mule. EHV-1 caused no clinically recognisable disease in 15 horses, 5 of which were introduced to the group 15 days before the first recumbent case. Three horses and the mule showed only fever and respiratory signs, while 14 horses developed neurological symptoms, leading to the death of 6 horses and persistent hind limb weakness in 3 horses.
The following minor corrections are recommended: line
7: Lake Tahoe
62: performed
Additional information needed in M&M:
- Time points of blood and nasal samples as well as no. of animals sampled ( or at least a reference to table 2).
- More detailed information on validation of assigning antibody levels to risk of developing EHV-1 disease should be given. Is this validated? How?
88 and 89: According to Table 1, the mule developed EHV-1: "...outbreak in 18 equids, while 15 horses (healthy cases)..."
Table 1: The order of the listed equidae should be grouped according to their status "healthy", "EHV-1" and "EHM". In addition, please mark (group) the five new and the dead horses, e.g. with superscripts.
Line 112 to 124 repeat information given in Table 2 and therefore should be deleted. A statement on the the essential information contained in table 2 would be desirable.
150: "While 8 EHM..."
150-154 and Table 3: More information is needed on the database, on which the IgG-based "risk scores" are defined and/or validated. An explanation should be given as to why EHV-1 antibodies may be related to risk of disease (knowing that there is no association with prevention of clinical disease). This can be included in the discussion.
184-187: Please explain why an isolation of new horses for 21 days will prevent shedding / introduction of EHV-1 to the residents
209-210: Since vestibular signs are relatively rarely caused by EHV-1 and other causes are possible, more detailed information on the no of affected animals and course is desirable
240-241: More information about how to measure mucosal anti-EHV-1 antibodies would be helpful
249-250: With the high EHM and fatality rate, the final sentence of the discussion seems inappropriate. Proposal: Overall, the observational study shows that the management of an EHM outbreak can be managed in the field without the results being worse than after treatment in clincal settings.
Author Response
7: Lake Tahoe
The correction was performed.
62: performed
We have corrected the misspelling.
Additional information needed in M&M:
- Time points of blood and nasal samples as well as no. of animals sampled ( or at least a reference to table 2).
The time points and number of animals tested has been added in M & M.
- More detailed information on validation of assigning antibody levels to risk of developing EHV-1 disease should be given. Is this validated? How?
The risk of disease assessment based on the level of anti-EHV-1 IgG and IgG 4/7 has been validated by the group of Cornell and has previously been published. We have added a reference to the M & M and also added the specific values in order to determine the level of risk.
88 and 89: According to Table 1, the mule developed EHV-1: "...outbreak in 18 equids, while 15 horses (healthy cases)..."
Many thanks for pointing out the issue between listing animals as horses, versus equids. Whenever the only mule was part of any descriptive parameters the authors have used the word “equids”.
Table 1: The order of the listed equidae should be grouped according to their status "healthy", "EHV-1" and "EHM". In addition, please mark (group) the five new and the dead horses, e.g. with superscripts.
Table 1 has been changed according to the reviewer’s suggestion.
Line 112 to 124 repeat information given in Table 2 and therefore should be deleted. A statement on the essential information contained in table 2 would be desirable.
The duplicate information has been removed.
150: "While 8 EHM..."
The typo was corrected.
150-154 and Table 3: More information is needed on the database, on which the IgG-based "risk scores" are defined and/or validated. An explanation should be given as to why EHV-1 antibodies may be related to risk of disease (knowing that there is no association with prevention of clinical disease). This can be included in the discussion.
Two previously published experimental studies have shown that systemic antibodies and mucosal antibodies can prevent infection. These two studies have set the basis for the EHV-1 risk assessment. Additional information pertaining to the parallel between systemic antibodies and protection has been added in the discussion.
- Goodman, L.; Wagner, B.; Flaminio, M.; Sussman, K.; Metzger, S.; Holland, R.; Osterrieder, K. Comparison of the efficacy of inactivated combination and modified-live virus vaccines against challenge infection with neuropathogenic equine herpesvirus type 1 (EHV-1). Vaccine 2006, 24, 3636–3645.
- Schnabel CL, Babasyan S, Rollins A, Freer H, Wimer CL, Perkins GA, Raza F, Osterrieder N, Wagner B. An Equine Herpesvirus Type 1 (EHV-1) Ab4 Open Reading Frame 2 Deletion Mutant Provides Immunity and Protection from EHV-1 Infection and Disease. J Virol. 2019 Oct 29;93(22):e01011-19.
184-187: Please explain why an isolation of new horses for 21 days will prevent shedding / introduction of EHV-1 to the residents
Isolation and monitoring of new arrivals at a facility is considered the standard of care to prevent possible transmission of contagious pathogens. The time frame of 21-28 days reflects 3-4 times the general incubation period for respiratory pathogens. Isolation and monitoring of new arrivals allow to first of all recognize clinical disease if a new arrival develops respiratory signs and second prevents spread if a new arrival is a transient subclinical shedder.
209-210: Since vestibular signs are relatively rarely caused by EHV-1 and other causes are possible, more detailed information on the no of affected animals and course is desirable
We agree with the reviewer that vestibular signs are a rare occurrence associated with EHM and have been sporadically reported during outbreaks. In order to prevent any confusion
240-241: More information about how to measure mucosal anti-EHV-1 antibodies would be helpful
The authors have added additional information regarding the measurement of mucosal anti-EHV-1 antibodies using a bead-based multiplex assay.
249-250: With the high EHM and fatality rate, the final sentence of the discussion seems inappropriate. Proposal: Overall, the observational study shows that the management of an EHM outbreak can be managed in the field without the results being worse than after treatment in clinical settings.
The sentence has been changed as suggested by the reviewer.
Reviewer 2 Report
Comments and Suggestions for Authors
Investigation of an Outbreak of Equine Herpesvirus-1 Myeloencephalopathy in a Population of Aged Working Horses
This manuscript describes an outbreak of equine herpesvirus-1 myeloencephalopathy (EHM) in a population of aged horses. The manuscript is well-written, and the detailed description of the results regarding EHV-1 detection by qPCR and the serological findings represents an important contribution to understanding the dynamics of EHM outbreaks. The manuscript is of interest to readers due to the emergence of EHM outbreaks in recent years in various countries, providing valuable and comprehensive information on the management of an outbreak in elderly leisure horses.
Below I have listed some recommendations:
Abstract:
In my opinion, the abstract contains too much data (results) that are not particularly essential, making this section overly dense and detracting from the key aspects of the outbreak. For instance, lines 16 to 18 could be revised to: “EHV-1 was detected by qPCR in the blood and/or nasal secretions of all EHM cases, three EHV-1 cases, and two healthy horses.” Another example is the final sentence, which does not add critical information and could be omitted to include more relevant data, such as the significant median viral load in EHM cases and the comparison between non-surviving and surviving EHM cases.
Materials and Methods:
Line 56: It would be interesting to add information regarding the conditions of “quarantine” for the 5 outside horses during those 9 days.
Line 67: Which anticoagulant was used?
Lines 80-81: Regarding the serological results for IgG and IgG 4/7, and considering the interesting findings presented, it would be essential to include an explanation of the risk score assignment within this section (currently explained later in Table 3). Doing so would significantly enhance the clarity and accessibility of the results, making them easier to follow.
This section is also lacking the information regarding the statistical methods for the comparisons amongst the three disease groups for total IgG and IgG 4/7, and also for other comparisons such as qPCR viral load results.
Results:
Line 144: should be double-checked, as a Kruskal-Wallis test should be used to compare acute serum anti-EHV-1 total IgG and IgG 4/7 values among the three disease groups if the data do not follow a normal distribution.
Discussion:
The first sentence should be slightly amended since the manuscript does not really provide much information regarding treatment options; it is however a highly valuable and complete source of information regarding the laboratory findings during an EHM outbreak.
The discussion is very interesting, reads well and provides the reader with relevant data. Findings have been properly referenced when previously observed in other studies. However, lines 230-233 would need to be revised, as the use of valacyclovir has been associated to a decrease in viremia in other articles:
- Pusterla N, Barnum S, Miller J, Varnell S, Dallap-Schaer B, Aceto H, et al. Investigation of an EHV-1 outbreak in the United States caused by a new H752 genotype. Pathogens. 2021; 10(6): 747.
- Maxwell LK, Bentz BG, Gilliam LL, Ritchey JW, Pusterla N, Eberle R, et al. Efficacy of the early administration of valacyclovir hydrochloride for the treatment of neuropathogenic equine herpesvirus type-1 infection in horses. Am J Vet Res. 2017; 78: 1126–1139.
- It was also suggested in: Velloso Alvarez A, Jose-Cunilleras E, Dorrego-Rodriguez A, Santiago-Llorente I, de la Cuesta-Torrado M, Troya-Portillo L, Rivera B, Vitale V, de Juan L, Cruz-Lopez F. Detection of equine herpesvirus-1 (EHV-1) in urine samples during outbreaks of equine herpesvirus myeloencephalopathy. Equine Vet J. 2024 May;56(3):456-463.
Overall, the manuscript is suitable for publication as a case report, provided the suggested amendments are made.
Author Response
Abstract:
In my opinion, the abstract contains too much data (results) that are not particularly essential, making this section overly dense and detracting from the key aspects of the outbreak. For instance, lines 16 to 18 could be revised to: “EHV-1 was detected by qPCR in the blood and/or nasal secretions of all EHM cases, three EHV-1 cases, and two healthy horses.” Another example is the final sentence, which does not add critical information and could be omitted to include more relevant data, such as the significant median viral load in EHM cases and the comparison between non-surviving and surviving EHM cases.
As suggested by the reviewer, the abstract has been condensed so to focus on the key aspect of the outbreak.
Materials and Methods:
Line 56: It would be interesting to add information regarding the conditions of “quarantine” for the 5 outside horses during those 9 days.
The 5 horses were kept in a separate paddock and had not direct contact to the resident horses. However, no specific biosecurity or monitoring protocols were performed during the 9 -ay separation period.
Line 67: Which anticoagulant was used?
Blood was collected in tubes with and without coagulant (EDTA). The missing information has been added.
Lines 80-81: Regarding the serological results for IgG and IgG 4/7, and considering the interesting findings presented, it would be essential to include an explanation of the risk score assignment within this section (currently explained later in Table 3). Doing so would significantly enhance the clarity and accessibility of the results, making them easier to follow.
This section is also lacking the information regarding the statistical methods for the comparisons amongst the three disease groups for total IgG and IgG 4/7, and also for other comparisons such as qPCR viral load results.
The risk of disease assessment based on the level of anti-EHV-1 IgG and IgG 4/7 has been validated by the group of Cornell and has previously been published. We have added a reference to the M & M and also added the specific values in order to determine the level of risk.
Statistical methods have also been added.
Results:
Line 144: should be double-checked, as a Kruskal-Wallis test should be used to compare acute serum anti-EHV-1 total IgG and IgG 4/7 values among the three disease groups if the data do not follow a normal distribution.
Yes, the one-way ANOVA and Kruskal-Wallis test are the same tests. We have placed more information as requested under Material and Methods.
Discussion:
The first sentence should be slightly amended since the manuscript does not really provide much information regarding treatment options; it is however a highly valuable and complete source of information regarding the laboratory findings during an EHM outbreak.
As requested by the reviewer, the “treatment options” has been removed.
The discussion is very interesting, reads well and provides the reader with relevant data. Findings have been properly referenced when previously observed in other studies. However, lines 230-233 would need to be revised, as the use of valacyclovir has been associated to a decrease in viremia in other articles:
- Pusterla N, Barnum S, Miller J, Varnell S, Dallap-Schaer B, Aceto H, et al. Investigation of an EHV-1 outbreak in the United States caused by a new H752 genotype. Pathogens. 2021; 10(6): 747.
- Maxwell LK, Bentz BG, Gilliam LL, Ritchey JW, Pusterla N, Eberle R, et al. Efficacy of the early administration of valacyclovir hydrochloride for the treatment of neuropathogenic equine herpesvirus type-1 infection in horses. Am J Vet Res. 2017; 78: 1126–1139.
- It was also suggested in: Velloso Alvarez A, Jose-Cunilleras E, Dorrego-Rodriguez A, Santiago-Llorente I, de la Cuesta-Torrado M, Troya-Portillo L, Rivera B, Vitale V, de Juan L, Cruz-Lopez F. Detection of equine herpesvirus-1 (EHV-1) in urine samples during outbreaks of equine herpesvirus myeloencephalopathy. Equine Vet J. 2024 May;56(3):456-463.
Indeed, the reviewer is absolute right that valacyclovir reduces viral loads in blood and nasal secretions. Our sentence reads “The use of valacyclovir in viremic horses had the positive effect of quickly reducing viremia and nasal shedding, as exemplified by the rapid decline of EHV-1 qPCR-positive horses on the second sample collection date (day 7 following onset of outbreak)”. We have added a sentence so to support the study results with previous literature.
Reviewer 3 Report
Comments and Suggestions for Authors
The manuscript describes an interesting study on clinical and laboratory findings, treatment and disease outcome of equine herpesvirus myeloencephalopathy in a population of aged working horses.
Overall, the manuscript is well-written, all sections are clearly presented and references appear to be sufficient.
The manuscript meets all the requirements for publication in Viruses.
Author Response
The authors thank the reviewer for the comments provided for this manuscript.